# Methodological Progress of Stereology in Cardiac Research and Its Application to Normal and Pathological Heart Development

**DOI:** 10.3390/cells11132032

**Published:** 2022-06-26

**Authors:** Christian Mühlfeld, Julia Schipke

**Affiliations:** 1Institute of Functional and Applied Anatomy, Hannover Medical School, 30625 Hannover, Germany; schipke.julia@mh-hannover.de; 2Biomedical Research in Endstage and Obstructive Lung Disease Hannover (BREATH), Member of the German Center for Lung Research (DZL), 30625 Hannover, Germany; 3Research Core Unit Electron Microscopy, Hannover Medical School, 30625 Hannover, Germany

**Keywords:** design-based stereology, cardiac development, cardiomyocyte number, intrauterine growth restriction, cardiac innervation, collagen deposition

## Abstract

Design-based stereology is the gold standard for obtaining unbiased quantitative morphological data on volume, surface area, and length, as well as the number of tissues, cells or organelles. In cardiac research, the introduction of a stereological method to unbiasedly estimate the number of cardiomyocytes has considerably increased the use of stereology. Since its original description, various modifications to this method have been described. A particular field in which this method has been employed is the normal developmental life cycle of cardiomyocytes after birth, and particularly the question of when, during postnatal development, cardiomyocytes lose their capacity to divide and proliferate, and thus their inherent regenerative ability. This field is directly related to a second major application of stereology in recent years, addressing the question of what consequences intrauterine growth restriction has on the development of the heart, particularly of cardiomyocytes. Advances have also been made regarding the quantification of nerve fibers and collagen deposition as measures of heart innervation and fibrosis. In the present review article, we highlight the methodological progress made in the last 20 years and demonstrate how stereology has helped to gain insight into the process of normal cardiac development, and how it is affected by intrauterine growth restriction.

## 1. Introduction

Imagine the following situation: During the planning of an immunohistochemical study you have the choice between two antibodies—the first one binds efficiently and specifically and costs $500, whereas the other is half the price but does not provide a specific staining at all. Which one would you choose? Of course, this is a rhetoric question. No serious scientist would prefer a method that does not work—even at low price.

Surprisingly, this seems to be different when it comes to quantifying structural features of a tissue, or of cells or organelles. One of the most frequently occurring examples is the question whether a certain condition causes a loss or an increase of cells. The most widely applied “method” to address this question is to look at histological sections and count the cells of interest, and divide it by the area used for counting. Thus, the number of cells is reported as “cells per mm²” or “cells per high power field”. Of course, this is an easy and quick way to obtain data since it does not cost much time. On the other hand, the data are inconclusive for two main reasons: (1) The data report a ratio which can be changed either by the numerator or the denominator. This problem is often referred to as the reference trap [1]. For example, when cells undergo hypertrophy, they will occupy a larger area in a tissue section and the number of profiles per area decreases. Therefore, the data do not answer the question whether there are more or less cells in the reference space. A conclusive answer can only be obtained by an estimate of the total number of cells in a defined reference space (e.g., the left ventricle of a heart). (2) The sectional profiles are treated as being representative of the whole cell. Due to the cutting process, each three-dimensional characteristic is reduced by one dimension. Thus, in a two-dimensional section, the volume of an object is represented by its area, its surface area by the one-dimensional boundary line, and the length of a structure by the number of times it is transected. As a zero-dimensional parameter, information about a number is not represented in a 2D section at all. Thus, an unbiased estimate is required to provide meaningful data about the volume, surface area, length or number (Figure 1A–D).

Design-based stereology, a branch of stochastic geometry, is the method of choice whenever quantitative morphological data are required to answer a biomedical research question. The theory and application of stereology have been addressed in various detailed review articles [2,3,4,5,6] and will not be repeated here. For cardiac research, a comprehensive review article [7], as well as a review article addressing the cardiac microcirculation [8], were published several years ago. The present article has two aims: (1) To give an update of recently developed or evaluated stereological methods; and (2) to review the scientific progress that has been made by the application of stereological methods. To provide a coherent illustration of the potential of stereology, the latter was restricted to the normal cardiomyocyte development and the effect of intrauterine growth restriction on this process.

## 2. Methodological Advances

In 1984, the first unbiased estimator of a number was described [9], the so-called disector. It is based on the counting of unique events (e.g., the tops or bottoms of objects) within a known volume between two thin sections (physical disector) or between two focal planes in a thick section (optical disector). The resulting numerical density (number of objects per unit tissue volume) is multiplied by the reference volume to obtain the total number of cells in the reference volume.

When mono-nucleated cells are the target of investigation, counting the number of nuclei provides an estimate of the number of cells. Cardiomyocytes, however, are known to contain one or more nuclei, and the fraction of cardiomyocytes with a different number of nuclei changes during development and during disease processes, and differs among species. The number of cardiomyocytes can therefore not directly be derived from the number of their nuclei. In 2005, Brüel and Nyengaard [10] described a two-step method that is based on estimating (1) the number of myocyte nuclei, and (2) the mean number of nuclei per myocyte. The first step requires the use of a disector, the second one requires immunohistochemically stained serial sections whose number depends on the size of the myocytes. After dividing the total number of nuclei by the mean number of nuclei per myocyte, the total number of cardiomyocytes is obtained. The most time-consuming step in this procedure is the estimation of the mean number of nuclei per cardiomyocyte. Various authors have adapted the method to facilitate this, e.g., by using confocal laser scanning microscopy of fluorescently stained thick sections [11] (Figure 2A–I), or by digesting a number of additional hearts and estimating the proportion of mono- and binucleated cells by flow cytometry [12]. When using the second method, one has to be aware that the mean number of nuclei is not obtained from the same hearts used for counting of the total number of nuclei, and that a small portion of cardiomyocytes contains three or even more nuclei. Moreover, flow cytometry cannot differentiate between nuclear ploidy and multinucleation; for this, a combination with imaging is required. Often, not only the number of cardiomyocytes itself but also the number of newly generated cardiomyocytes is of interest. In this regard, a limitation of the stereological number estimation is that it only provides information for a given point in time, but does not provide a dynamic picture of the process that has led to this number. For this purpose, an estimate of cardiomyocyte proliferation is required. Very recently, Sampaio-Pinto et al. [13] proposed a stereological method to analyse the proliferation of cardiomyocytes by estimating cardiomyocytes after a 5-ethynyl-2′-deoxyuridine injection. This method is of particular interest because it adds a dynamic component to cardiomyocyte number estimation.

In addition to these stereological approaches, digestion of hearts and counting of cells by flow cytometry are also used [14]; however, it needs to be mentioned that studies comparing the estimated number of pulmonary alveolar epithelial cells obtained by stereology or flow cytometry have shown that the cytometric approach significantly underestimates the total number of cells [15,16].

Additional progress has been made in the past years concerning the innervation of the heart. In 2010, Mühlfeld et al. [17] established a stereological method to quantify the total length of axons innervating the heart. The method is based on counting the number of immunohistochemically stained nerve fiber profiles using light microscopy, and on estimating the mean number of axons per nerve fiber profile using transmission electron microscopy (TEM). From these estimations, the total length of axons innervating the left ventricle was calculated. The original method was described with an antibody against the pan-neuronal marker PGP9.5; however, using antibodies against different types of neurons has the potential to distinguish, e.g., between neurons of the sympathetic or parasympathetic nervous system. As a less efficient modification, the length of axons was also estimated solely at the TEM level, thus enabling the use of archived material provided the material was randomly sampled from the reference volume [18].

One of the most frequently estimated morphometric parameters in cardiac research is the volume fraction of collagen fibers to detect the deposition of fibrotic tissue between cardiomyocytes. Although stereology is the gold standard to perform these estimates, other quantification tools such as semi-automated analyses of digital microscopic images of cardiac tissue stained by picrosirius red have led to similar results to stereology but requiring less time [19]. A similarly strong correlation between digital image analysis and stereology was observed in human bioptic specimens by Daunoravicius et al. [20] who also showed that both techniques were superior to visual scoring by a pathologist alone. Schipke et al. [21] compared stereological estimates at the light and TEM level to digital image analysis. While TEM enabled the distinction between different locations of fibrotic tissue, it was rather time-consuming. The light microscopic data depended on the thickness of the section in use; thus, with increasing section thickness, the amount of fibrotic tissue was overestimated. Here, the digital semi-automatic estimation correlated well with the manually determined volume fraction. It is important to note that each of these methods leads to a volume fraction that carries the potential of ignoring the reference trap. A problematic issue that aggravates the use of the ratio is the common use of paraffin-embedded tissue that shrinks during the embedding procedure in a non-predictable fashion [22], i.e., the higher the water content of the sample, the greater the shrinkage. Thus, differences of the volume fraction of collagen between two experimental groups may be strongly influenced by the degree of shrinkage of the whole sample during embedding. To avoid this problem, tissue embedding procedures that minimize shrinkage, such as osmium tetroxide/uranyl acetate/glycol methacrylate [22], should be utilized for collagen quantification analyses in future studies. 

It is no secret that stereology requires more time than other more simplistic morphometric methods, but this cannot be a justification for using biased methods. Although stereological methods still rely on manual counting procedures to a great extent, their use has been made much easier by free or commercially available stereology software. Similarly, the use of automated slide scanners has greatly improved the efficiency of investigation. Nevertheless, it can be expected that progress in non-destructive imaging, image analysis, and machine learning will further increase the efficiency of stereological methods and decrease the amount of time spent by researchers doing manual counting. However, even with such future progress, researchers will still need to be able to evaluate the plausibility and validity of automatically generated results, possibly by comparison with manually acquired results from the same sample. 

## 3. Developmental Physiological Growth and Cardiomyocyte Number

A broad understanding of the physiological cardiac development is essential to recognize and assess fetal and postnatal growth defects that might prime the heart for injury or disease in later life. Moreover, there is the hope that decoding of molecular mechanisms responsible for cardiac proliferation and differentiation could enable new therapeutic strategies that promote endogenous regeneration of the heart under pathologic conditions like cardiac infarction or heart failure. The question of the timing and the termination of cardiomyocyte proliferation during cardiac growth is still a matter of debate. Mollova et al. [23] reported a 3.4-fold increase in cardiomyocyte numbers between left ventricles of humans aged 1 year (1.1 × 10^9^) and 20 years (3.7 × 10^9^), indicating that cardiomyocyte proliferation contributes to physiological heart growth. This result is in stark contrast to Bergmann et al. [24], who stated that the final number of left ventricular cardiomyocytes (3.2 × 10^9^) was already reached 1 month after birth, and thus the human heart increases in size by hypertrophy of existing cardiomyocytes. Both studies utilized an optical disector for the estimation of the cardiomyocyte nuclei number [9,10], whereas only Bergmann et al. [24] included the cardiomyocyte-specific nuclear marker pericentriolar material 1 (PCM-1). However, it is questionable whether this difference can solely account for the discrepant results. Nevertheless, the use of a cardiomyocyte-specific nuclear marker is recommended in hearts of neonatal or juvenile individuals because at that age it may be difficult to distinguish between cardiomyocyte, endothelial cell or fibroblast nuclei. Although this becomes less difficult with increasing age, it may still facilitate the estimation in older organisms. Besides cardiomyocyte number, both studies include further experiments to underpin their findings. Mollova et al. [23] found cardiomyocytes positive for M-phase and cytokinesis markers up to an age of 20 years, albeit to a far lower range compared to the first year of life. In addition, 14C dating of cardiomyocytes indicates cardiomyocyte proliferation, especially in the first decade of life; however, according to Bergmann et al. [24], this proliferation is thought to replace existing cells, rather than to generate additional cardiomyocytes.

In the mouse heart, cardiomyocyte proliferation is thought to end after the first postnatal week [25,26], when myocytes become binucleated and terminally differentiated [27,28]. This view was challenged by Naqvi et al. [29], postulating a proliferative burst of cardiomyocytes at postnatal day 15 yielding an increase in myocyte numbers by 40%. As a limitation of this study, the cardiomyocyte number was assessed by cardiomyocyte isolation by Langendorff perfusion, differentiation from other cell types by cytoplasmic size, and counting with a hemocytometer. This approach possibly induced a quantification bias due to age-dependent different isolation efficiencies. As a direct response, Alkass et al. [25] revised this issue by several methods, including stereological cardiomyocyte quantification, visualization of DNA replication by EdU, BrdU and 15N-thymidine, as well as detection of cell proliferation by Ki-67, and found no evidence for myocyte proliferation between day 13 and 100. On the other hand, endurance training of rats between postnatal week 5 and 9 increased left ventricular cardiomyocyte numbers by 40% compared to sedentary animals, indicating an inducible proliferative potential during this juvenile stage [30]. While treadmill running during adolescence (week 11–15) still increased cardiomyocyte numbers by 20%, it had no effect on myocyte numbers in adult rats trained between weeks 20 and 24, thus the exercise-induced proliferation capacity seems to be lost during adulthood. Additionally, the administration of growth hormone was shown to stimulate cardiomyocyte proliferation, even in adult (3-month-old) rats, as shown by significant increases in cardiomyocyte numbers, as well as Ki-67 positive myocytes [31]. In contrast, in mice that had voluntary access to a running wheel for four weeks starting at the age of eight weeks, the cardiomyocyte number did not increase compared with their sedentary controls [32].

The transition from the proliferative to the terminally differentiated, non-proliferative phenotype of cardiomyocytes is accompanied by a multinucleation and/or onset of polyploidy [24,25]. This transition is accomplished by a premature cell cycle exit, either after DNA duplication during the S phase before karyogenesis (resulting in polyploidy), or after karyogenesis without successful cytogenesis (resulting in polynucleation) [33]. Cardiomyocytes in the mouse heart are mostly multinucleated [25,34], whereas human adult cardiomyocytes harbor usually one polyploid nucleus [23,24]. The number of nuclei per myocyte can be either assessed by stereological analysis of tissue sections [24], or by cytometry analysis of isolated myocytes [23]. Both methods provided similar estimates for the human heart; approximately 70% of mononucleated cardiomyocytes throughout the lifetime [23,24]. Interestingly, other mammalian cell types are also polyploid, i.e. hepatocytes, and are still able to proliferate and thus to regenerate cell losses [35,36]. Consequently, polyploidy is not necessarily associated with a permanent exit from the cell cycle like it seems to be the case in cardiomyocytes. Importantly, it was recently shown that mononucleated and binucleated cardiomyocytes of mouse hearts show different molecular patterns with regard to their regenerative potential, which highlights the necessity to distinguish between mono- and multinucleated cardiomyocytes in quantitative studies [37].

Thus, the timing and the regulation of cardiomyocyte proliferation is still a matter of debate, and future studies are needed to better understand induction and arrest of cell division in the heart, the underlying molecular cues, and the importance of multinucleation and ploidy for these processes.

## 4. Intrauterine Growth Restriction and Cardiomyocyte Number

During fetal development, changes of the intrauterine milieu can lead to intrauterine growth restriction (IUGR), which increases the risk for various diseases in childhood and adult life [38]. Known causes of IUGR are maternal malnutrition or cigarette smoking during pregnancy, or placental insufficiency, among others. Neonates born after IUGR are of smaller size and have a higher risk of suffering from ischemic heart disease and heart failure in later life [39,40]. Therefore, several stereological studies have addressed the consequences of experimental IUGR on cardiac characteristics, such as cardiomyocyte number or vascularization. In addition, studies have tested how IUGR translates into cardiac alterations in adult life. Experimental models include maternal hypoxia and nutrient restriction [41], modification of maternal dietary protein-content [12,42], decreased placental perfusion to induce fetal hypoxemia [43,44], and placental restriction by surgical removal of endometrial-placental attachment sites [45].

As mentioned above in the paragraph on the physiological growth of the heart, the proliferative capacity of cardiomyocytes is thought to decrease shortly after birth. Thus, a low number of cardiomyocytes at birth may result in a smaller number of cardiomyocytes in adult life, and may thus be a lifelong burden. Several studies have estimated the number of cardiomyocytes in newborns after IUGR. Corstius et al. [12] used maternal protein restriction to induce IUGR in rats. The study was carried out on the hearts of female offspring sacrificed on the day of birth. Interestingly, in this study, the whole heart, including the atria and ventricles, was included, which is quite uncommon in cardiac research where in most cases the left ventricle including the interventricular septum is analysed. The number of cardiomyocyte nuclei was estimated using an optical fractionator design [46], and the fraction of binucleated cells were assessed by light microscopy of enzymatically isolated cardiomyocytes taken from a different set of hearts. They showed that both the number of cardiomyocyte nuclei and cardiomyocytes was significantly lower in the low protein diet group (1.15 × 10^7^) compared with the normal diet group (1.43 × 10^7^). Thus, lower body and heart weight at birth due to maternal protein restriction was directly associated with a smaller number of cardiomyocytes. Schipke et al. [44] ligated uteroplacental arteries to induce IUGR in rabbits and obtained similar results, indicating that both decreased perfusion and maternal diet can affect the heart in a similar manner, and that the reduction of cardiomyocyte number occurs in IUGR in different species. Schipke et al. [44] used a physical disector design on epoxy resin-embedded samples to estimate the number of cardiomyocyte nuclei, and confocal laser scanning microscopy to estimate the mean number of nuclei per cardiomyocyte. The study differentiated between the left ventricle (incl. the interventricular septum) and the right ventricle (free wall) and showed that the number of cardiomyocytes was significantly decreased in the left ventricle but not in the right ventricle, suggesting a stronger effect of IUGR on the left than on the right ventricle. This was supported by the additional analysis of the capillaries, which were also more affected in the left than in the right ventricular myocardium.

Various studies have investigated whether these effects cause long-term effects of IUGR on cardiomyocyte number in different species and at different periods after birth. Lim et al. [42] used the rat model of maternal protein restriction and analysed the number of cardiomyocytes at four weeks of age in male and female offspring. Here, the data refer to both ventricles including the free wall and interventricular septum. However, only the number of cardiomyocyte nuclei was counted in a fractionator design, whereas the mean number of nuclei per cardiomyocyte was set as two under the assumption that all cardiomyocytes are binucleated at 4 weeks of age. Thus, the reported cardiomyocyte number was calculated as the number of cardiomyocyte nuclei divided by two. The obtained data did not show a significant difference between low and normal protein diet in male or female rats, but the number of cardiomyocytes was lower in the female than in the male rats and showed a close correlation with heart volume. The study suggests that the postnatal rat heart catches up with the effects of IUGR induced by protein restriction.

On the contrary, rodents are born in a more immature state than other mammals which makes it possible that, in other species, the IUGR-induced reduction of cardiomyocyte number continues into adulthood. This question was addressed by Vranas et al. [45] using a surgical model of placental restriction in male ovine offspring at one year of age. Here, the data were reported for the left ventricle including the interventricular septum. The authors estimated the number of cardiomyocyte nuclei in a fractionator design and performed confocal microscopy on paraffin sections stained for wheat germ agglutinin and DAPI to estimate the mean number of nuclei per cardiomyocyte. Although the total number of cardiomyocytes did not differ significantly between the control and the IUGR group, it is possible that the missing significance is caused by the low number of investigated animals (*n* = 5 in each group), particularly since the birth weight of the animals showed a great variation. However, the most interesting finding of the study was that the number of cardiomyocytes in the adult left ventricle correlates with body weight at birth.

Botting et al. [41] directly compared the effects of two different models of experimental IUGR on cardiomyocyte number in adult guinea pigs (119 days after birth), viz. maternal hypoxia (12% oxygen in ambient air) and food restriction (18–33% reduction compared to controls). The authors further distinguished between male and female offspring; however, this reduced the number of animals in each group to *n* = 4–5. The number of cardiomyocyte nuclei was estimated in the left ventricular free wall using an optical disector design, and the mean number of nuclei per cardiomyocyte was estimated from serial thin sections, as described originally by Brüel and Nyengaard [10]. Their results demonstrated striking differences between male and female animals, which highlights the need to distinguish between sexes in such studies. First of all, the cardiomyocyte number was higher in control females than in male offspring, and additionally, the number of cardiomyocytes was decreased in offspring of hypoxic but not calorically restricted mother animals. In contrast, IUGR had no effect on cardiomyocyte number in male offspring in either experimental model.

Taken together, despite the use of different methodological variants to induce IUGR and to estimate the number of cardiomyocytes, there is clear evidence that IUGR decreases fetal cardiomyocyte development in various species and leads to a smaller number of cardiomyocytes at birth. Whether this effect persists until adulthood or can be caught up during postnatal development requires further research. It is likely that the answer to this question depends on the species, sex and cause of IUGR [12,41,42,43,44,45]. 

## 5. Concluding Remarks

Stereology has a long tradition in biomedical research and continues to be an important tool to address questions whose answers require robust and reliable quantitative morphological data. Recent years have seen advances in quantification of cell number, innervation and fibrosis. In cardiac research, the introduction of an unbiased method to quantify the number of cardiomyocytes has led to an increased frequency of studies that address the normal or pathological development of the heart. Such studies have challenged the dogma that cardiomyocytes lose their capacity of proliferation within a short period after birth. They have also shown that IUGR decreased the number of cardiomyocytes in newborns, and that the mode of IUGR, the species, or sex of the animals influences the effect of IUGR and probably its long-term consequences. The present review article has shown the importance of using unbiased stereological methods for obtaining morphometric information in cardiac research. We hope that the new knowledge gained by these methods will encourage scientists in cardiac research to further implement stereology in their study designs. 

## Figures and Tables

**Figure 1 cells-11-02032-f001:**
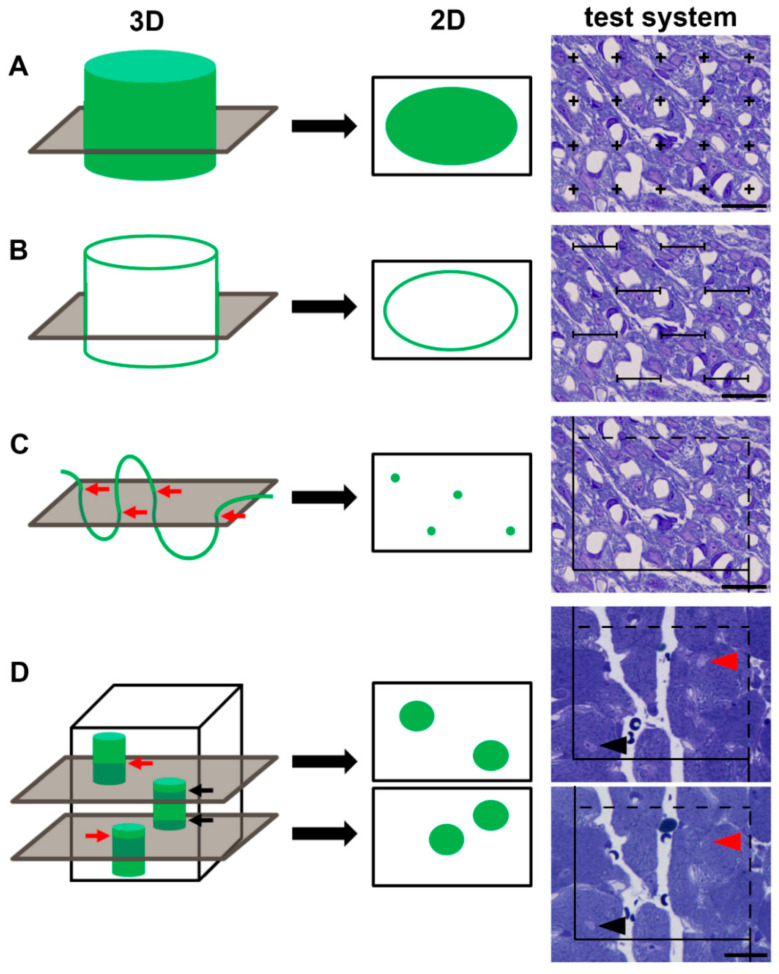
Structural parameters, their representation in 2D sections, and their estimation by stereological test systems. (**A**) A 3-dimensional (3D) volume appears as a 2-dimensional (2D) area in histological sections, and can be quantified by point probes; (**B**) a surface area of a 3D object appears as a 2D boundary line in histological sections, and can be quantified by line probes; (**C**) the length of a 3D object is represented by 2D profiles or transects (red arrows) in histological sections, and can be quantified by plane probes such as counting frames; (**D**) the number of objects is not represented in one histological section, but as particle tops or ends in a volume between two 2D sections, and can be quantified by plane probes and counting particles visible in one section, but not in the other (red arrows, red arrowheads). Objects transecting both sections appear as profiles on both sections and are not counted (black arrows, black arrowheads). Scale bar: 20 µm.

**Figure 2 cells-11-02032-f002:**
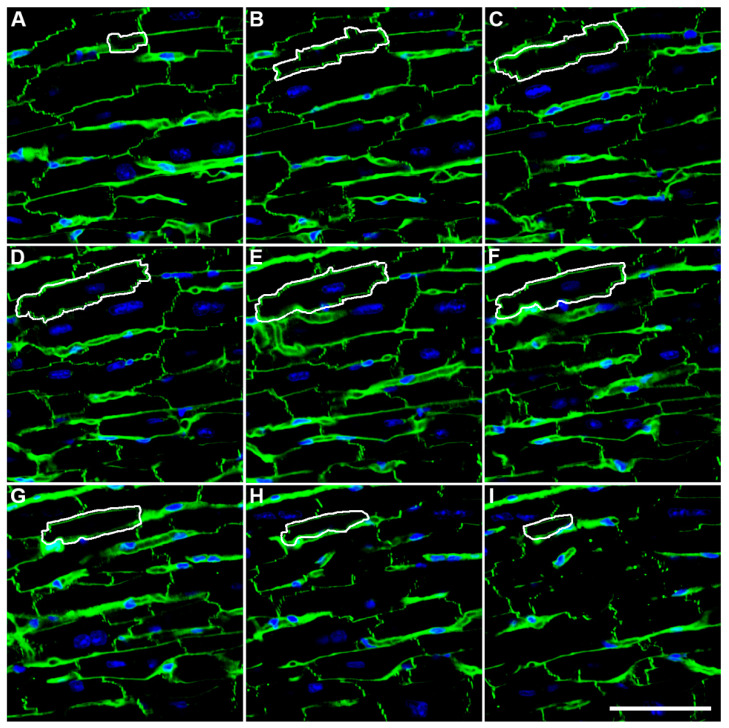
Estimation of the mean number of nuclei per cardiomyocyte. Consecutive optical sections (**A**–**I**) of a confocal microscopy z-stack are shown ((**A**): top, (**I**): bottom), which were used to follow up cardiomyocytes (one example indicated by a white line) and to determine the number of nuclei present in these cells (one nucleus in the indicated myocyte, visible in (**D**–**F**)). Shown are merged channels, with green = cadherin and wheat germ agglutinin, and blue = nuclei. Scale bar: 50 µm. Reprint with permission from ref. [11], Copyright 2014 John Wiley and Sons.

## Data Availability

Not applicable.

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
