# Peer review of "Methodological Progress of Stereology in Cardiac Research and Its Application to Normal and Pathological Heart Development"

_cells, 2022, doi:10.3390/cells11132032_

Round 1

Reviewer 1 Report

See attached file.

Reviewer 2 Report

Current review article collected the methodological progress made in the last years to demonstrate how stereology has been applied into the process of normal cardiac development in association to the intrauterine growth restriction (IUGR). Please conduct the concerns below.

1.      For cardiac research, two established reports [7, 8] were not challenged by others? Please check it in clear.

2.      Quantification of the estimated morphometric parameters has been concerned in many studies. As mentioned in line 149, the common use of paraffin which made the differences of the volume fraction of collagen between two experimental groups. How to improve this problem?

3.      The regulation of cardiomyocyte proliferation seems still unclear. Limitations of stereology used in it were not conducted in clear.

4.      The IUGR has been targeted in the review article. However, IUGR decreases fetal cardiomyocyte development in various species and leads to a smaller number of cardiomyocytes at birth without reference(s).

5.      Variations depending on species, sex and cause of IUGR need the evidence or reference(s) to support.

6.      In line 314, the showed sentence is hard to follow. Novelty of current review article was not indicated in the conclusion.

Round 2

Reviewer 1 Report

I am happy with the revised version of the manuscript. Appropriate corrections have been made. Best regards, Kanar Alkass